# scientific report

# Loss of iron triggers PINK1/Parkin-independent mitophagy

*George F.G. Allen[1], Rachel Toth[1], John James[2] & Ian G. Ganley[1+]*

[1]MRC-Protein Phosphorylation and Ubiquitylation Unit, and [2]Cell Signalling and Immunology, College of Life Sciences, University of Dundee, Dundee, UK

In this study, we develop a simple assay to identify mitophagy inducers on the basis of the use of fluorescently tagged mitochondria that undergo a colour change on lysosomal delivery. Using this assay, we identify iron chelators as a family of compounds that generate a strong mitophagy response. Iron chelation-induced mitophagy requires that cells undergo glycolysis, but does not require PINK1 stabilization or Parkin activation, and occurs in primary human fibroblasts as well as those isolated from a Parkinson's patient with Parkin mutations. Thus, we have identified and characterized a mitophagy pathway, the induction of which could prove beneficial as a potential therapy for several neurodegenerative diseases in which mitochondrial clearance is advantageous.

Keywords: autophagy; iron/mitophagy; PINK1; Parkin

## INTRODUCTION

Mitochondrial damage is associated with many human diseases and maintenance of a viable pool of mitochondria is considered fundamental to cell function [1]. Dysfunctional mitochondria produce reactive oxygen species (ROS) and so it is important for the cell to sequester and remove them before damage occurs. Recently, the molecular detail of how damaged mitochondria are segregated and degraded has begun to be elucidated. This process, termed mitophagy, involves mitochondrial engulfment by autophagosomes and delivery to lysosomes [2]. Mitophagy is thought to be important in long-lived and slow dividing cells, such as neurons or cardiomyocytes that cannot dilute mitochondrial DNA mutations through cell division. Hence, mitophagy is linked to ageing, heart disease and neurodegeneration [3–6]. Therefore, a need exists to identify pathways that can regulate this process.

Little is known about signals that trigger mitophagy and how the autophagic machinery is engaged. The role of mitophagy in Parkinson's disease has been an area of interest following observations that the protein kinase PINK1 and E3 ubiquitin ligase Parkin, both mutated in early onset forms of Parkinson's disease, act to induce mitophagy on mitochondrial membrane depolarization [7]. This implies, at least in patients harbouring mutated PINK1 or Parkin, impaired removal of mitochondria might contribute to neuronal death. Early studies were dependent on Parkin overexpression and recent evidence suggests endogenous Parkin is not sufficient to induce mitophagy, or at least not at a level that could be demonstrated with currently available techniques [8]. In contrast, the increased half-life of mitochondrial proteins in Parkin-deficient flies correlated with that in autophagy-deficient flies, indicating involvement of endogenous Parkin in mitophagy [9].

To avoid Parkin overexpression problems and to uncover pathways that could be targets for disease therapy, we developed a fluorescence assay to specifically monitor mitophagy. Using this assay, we screened compounds known to disrupt mitochondrial function or induce autophagy and identified iron chelation as a strong PINK1/Parkin-independent activator of mitophagy.

## RESULTS AND DISCUSSION

### A chemical screen for mitophagy inducers

We developed a simple and robust cell-based assay to specifically monitor lysosomal turnover of mitochondria. The assay relies on differences in pKa of green fluorescent protein (GFP) and mCherry and is adapted from a method developed for general autophagy [10]. The assay consists of cells expressing a tandem mCherry–GFP tag attached to the outer mitochondrial membrane localization signal of the protein FIS1 (residues 101–152). Under normal conditions, mitochondria fluoresce red and green. Upon mitophagy, mitochondria are delivered to lysosomes where the low pH quenches the GFP signal but not mCherry. The result is that a portion of mitochondria form punctate structures and fluoresce red only, with the degree of mitophagy calculated by their increase (Fig 1A). The construct was expressed in human U2OS osteosarcoma and SH-SY5Y neuroblastoma cells and found to be almost exclusively mitochondrial on the basis of co-localization with endogenous ATP synthase (Fig 1B) or MitoTracker (supplementary Fig S1A online). Consistent with the predicted long half-life of mitochondria in many cell types [11],

[1]MRC-Protein Phosphorylation and Ubiquitylation Unit, and
[2]Cell Signalling and Immunology, College of Life Sciences, University of Dundee, Dundee DD1 5EH, UK
[+]Corresponding author. Tel: +01382 388905; Fax: +01382 223778;
E-mail: i.ganley@dundee.ac.uk

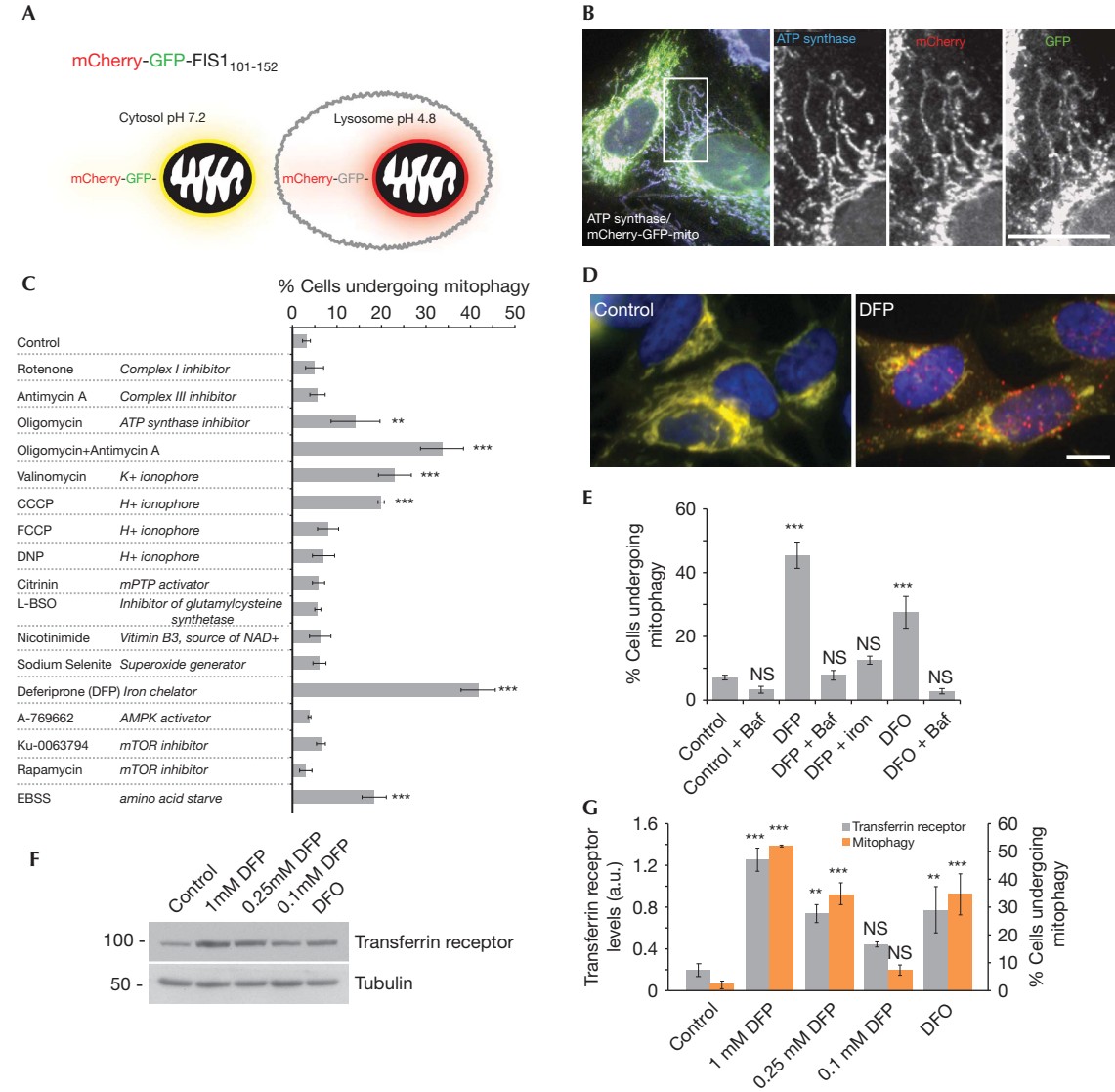

**Fig 1 | A chemical screen for mitophagy inducers.** (**A**) Diagram of tandem-tagged mCherry-GFP-FIS1$_{101-152}$-based mitophagy assay. (**B**) Co-localization of mCherry-GFP-FIS1$_{101-152}$ in U2OS cells with ATP synthase. (**C**) Screen for mitophagy inducing conditions using the tandem-tag mitophagy assay in SH-SY5Y cells. All treatments for 24 h. For this experiment, quantitation of mitophagy was performed with counter blinded to condition. (**D**) Example micrographs from cells expressing the mitophagy construct treated under control conditions or with 1 mM DFP for 24 h. (**E**) Results of mitophagy assay in SH-SY5Y cells following treatment with 1 mM DFP or deferoxamine for 24 h, 340 μM ferric ammonium citrate (iron) was preincubated with 1 mM DFP for 10 min before addition to cells. 100 nM bafilomycin A1 was added for final 4 h of incubation. (**F**) Representative blot and (**G**) Quantitation of transferrin receptor protein levels (left axis) and results of tandem-tag mitophagy assay (right axis) from SH-SY5Y cells treated with DFP at indicated concentration or 1 mM DFO for 24 h. All results are from 3–4 independent experiments. All quantitative data are mean ± s.e.m. Scale bars, 10 μm. **$P < 0.01$, ***$P < 0.001$, NS, not significant. Baf, bafilomycin A1; DFO, deferoxamine; DFP, deferiprone; GFP, green fluorescent protein. Source data for this figure is available on the online supplementary information page.

little mitophagy was observed under control conditions. To find inducers of mitophagy, we used the assay to screen compounds that affect aspects of mitochondrial physiology or have been reported to induce mitophagy (Fig 1C). Treatment with several of these compounds for 24 h led to mitophagy as indicated by red-alone puncta, corresponding to the mitochondrial tag in acidic lysosomes, confirmed by LysoTracker staining (supplementary Fig S1B,C online). We found mitophagy was induced to varying levels by the H$^+$ ionophore CCCP, the K$^+$ ionophore

valinomycin and a combination of the complex III inhibitor antimycin A and the ATP synthase inhibitor oligomycin A. All have previously been implicated in PINK1/Parkin activation [7,8,12]. We note the compounds did not result in complete mitochondrial removal, as has been observed in some cases on Parkin overexpression. This is unsurprising as it is hard to predict a scenario where complete clearance of mitochondria from a cell would be beneficial. From our screen, the strongest mitophagy inducer was the iron chelator deferiprone (DFP–see Fig 1D). As

iron homoeostasis is vital for cell function, we further characterized the effects of this chemical. With DFP, we found bafilomycin A1 treatment, to raise the pH of lysosomes, caused a loss of red-only puncta, confirming delivery of the mitochondrial tag to lysosomes and that the assay responds dynamically to lysosomal pH (Fig 1E). To directly implicate iron, we found that DFP pretreatment with $Fe^{3+}$ blocked its ability to stimulate mitophagy (Fig 1E). Five structurally distinct iron chelators, including deferoxamine, also stimulated mitophagy (Fig 1E–G, supplementary Fig S1D online). To strengthen the link between iron chelators, iron levels and mitophagy, we looked at transferrin receptor levels, which are increased on intracellular iron depletion [13]. We found a close correlation between mitophagy and transferrin receptor levels in response to iron chelator dose or type (Fig 1F,G), supporting a role for iron loss in mitophagy. Additionally, the degree of mitophagy peaked at ~24 h of treatment with 1 mM DFP (supplementary Fig S1E,F online).

## Loss of iron induces mitophagy

To validate our assay and a role for iron, we used methods independent of the tandem mCherry–GFP mitochondrial tag. Immunofluorescence showed DFP treatment caused an increase in LC3 puncta (autophagosome) formation in SH-SY5Y cells; ~45% of these structures co-localized with COXIV, a complex IV component (Fig 2A,B). Importantly, this co-localization increased to 65% after bafilomycin addition. This contrasts to starvation-induced autophagy that while inducing a similar number of autophagosomes (supplementary Fig S2A online), did not result in significant LC3-COXIV co-localization (Fig 2A,B). Thus iron chelation specifically induces mitophagy rather than general autophagy. The ~10% of co-localizing structures under control conditions might represent basally forming autophagosomes as evidence suggests they require mitochondrial membrane or mitochondria to mark the initiation site [14,15]. It is important to note that iron chelation did not grossly disrupt mitochondrial localisation, although the majority of fragments co-localizing with LC3 were separate from the network (Figs 2A and 3A, supplementary Fig S1F online).

Immunoblotting showed protein from each mitochondrial compartment decreased ~50% following DFP treatment (Fig 2C,D). This was prevented with bafilomycin indicating the decrease was dependent on lysosomal degradation. The exception was mitofusin2 (MFN2) that is degraded via the proteasome before mitophagy ([16] and supplementary Fig S2B online). As with immunofluorescence (supplementary Fig S2A), DFP treatment results in increased LC3-II autophagic flux. Expression of the autophagy adaptor and substrate SQSTM1/p62 increased following mitophagy induction and the amount of p62 was further enhanced with bafilomycin treatment. This suggests a role for p62 in DFP-induced mitophagy, although work is needed to clarify this. As a biochemical measure of mitochondrial abundance, citrate synthase activity was determined. DFP treatment reduced citrate synthase activity to 52% of control and lysosomal dependency was demonstrated with bafilomycin (Fig 2E). Citrate synthase activity loss on DFP treatment could be prevented in HeLa cells by depleting core autophagy genes ATG5 or Beclin1 (Fig 2F). Finally, we employed electron microscopy as the gold standard to confirm mitophagy (Fig 2G,H). DFP treatment resulted in a sixfold increase in mitochondria surrounded by multiple limiting membranes (autophagosomes) or by single membranes (autolysosomes). Together, the data clearly indicate mitophagy is induced by loss of iron.

## Effects of iron chelation on mitochondrial function

A major question concerns how loss of iron triggers mitophagy. Iron chelation is known to stabilize the oxygen responsive transcription factor HIF1α, through inhibition of the proline hydroxylase involved in its degradation. Hypoxia has also been shown to induce mitophagy via HIF1 [17]. We tested conditions previously reported to stabilize HIF1α (including hypoxia, proline hydroxylase inhibitors such as DMOG and $CoCl_2$, and iron chelation) with our assay. All conditions that significantly stabilized HIF1α also induced mitophagy (supplementary Fig S3A,B online). However, high levels of HIF1α did not always correlate with high levels of mitophagy (especially in the case of hypoxia itself and the iron chelator Dp44mT), implying HIF1α stabilization might be important but not sufficient to drive mitophagy on iron chelation. The mitochondria-localized HIF1-dependent gene *BNIP3* has previously been reported to act as a mitophagy receptor and have a role in autophagy induction [18,19]. In spite of lysosomal turnover after DFP (Fig 2C), BNIP3 small interfering RNA (siRNA) depletion did not influence mitophagy suggesting it is dispensable in this instance (supplementary Fig S3C,D online).

We reasoned as the autophagy is selective for mitochondria, iron loss might impair mitochondrial function that in turn signals for mitophagy. Mitochondria produce iron–sulphur clusters and haem groups required for many mitochondrial and cytosolic enzymes, including all four complexes of the respiratory chain. Therefore, loss of iron could disrupt respiration. We analysed mitochondrial function using MitoTracker, a dye that requires complex III activity and membrane potential to accumulate in mitochondria [20]. Iron chelation did not cause a loss in MitoTracker staining, which is in contrast to the almost complete loss on oligomycin/antimycin combination or CCCP treatment (Fig 3A, supplementary Fig S4A online). This indicates that mitophagy induction by oligomycin/antimycin or CCCP is potentially different from iron chelation. There are MitoTracker-negative mitochondrial structures on iron chelation and possibly, these are undergoing mitophagy, although we are unable to determine whether this is a cause or consequence of mitophagy (supplementary Fig S4A online). Next, we measured oxygen consumption and found basal and maximal respiration were decreased following only 4 h of iron chelation, that is, before mitophagy initiation (Fig 3B, supplementary Fig S1F online). By 24 h, oxygen consumption was essentially abolished, even though cells retain around half of their mitochondria (Fig 2). Despite respiration loss, ATP levels were maintained after DFP treatment for 24 h (Fig 3C). This is indicative of a switch in ATP production from oxidative phosphorylation to glycolysis, reminiscent of the Warburg effect in some cancer cells. To investigate a mitophagy role in this potential metabolic switch, we cultured cells in media containing galactose instead of glucose to force the cells to depend on oxidative phosphorylation for ATP production. Galactose conditions blocked iron chelator-induced mitophagy indicating glycolytic metabolism is essential (Fig 3D). A similar situation has been observed for Parkin mitophagy [21] and in yeast [22]. Together these data imply the metabolic energy requirements of the cell are fundamental in determining the

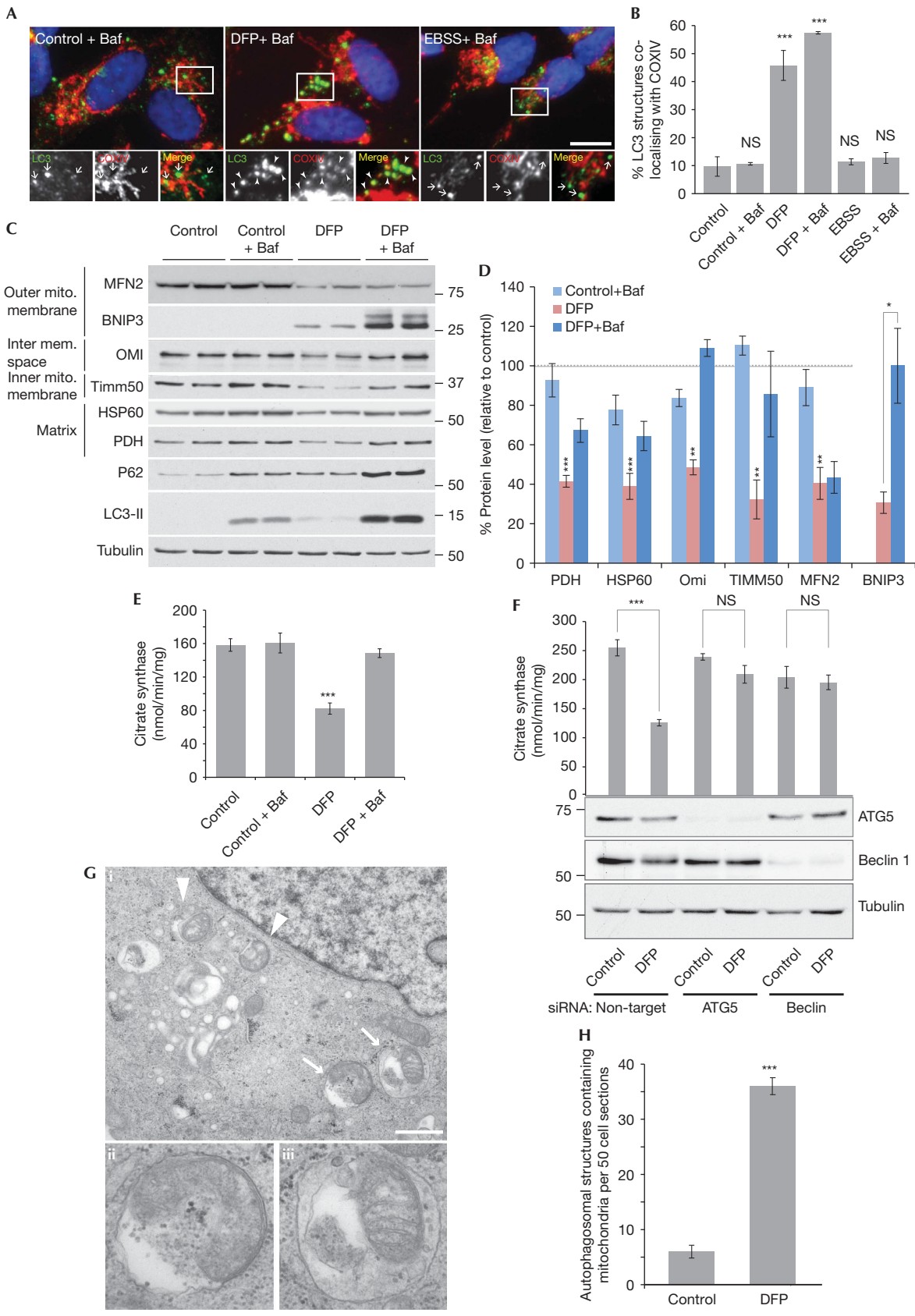

**Fig 2 | Loss of iron induces mitophagy. (A)** Representative images and **(B)** quantitation of SH-SY5Y cells stained with LC3 and COXIV antibodies, 1 mM DFP treatment for 16 h, EBSS (amino-acid starvation) for 2 h and 50 nM bafilomycin A1 for final 2 h of treatment. Arrows indicate non co-localized LC3 puncta and arrowheads co-localized puncta. Scale bar, 10 μm. **(C)** Example immunoblot and **(D)** Quantitation of mitochondrial proteins and autophagy markers in SH-SY5Y cells treated with 1 mM DFP for 24 h, 50 nM bafliomycin A1 was added for final 16 h of treatment. Data relative to control condition - dotted line represents control value (100%). **(E)** Citrate synthase activity following 1 mM DFP treatment for 24 h, 50 nM bafilomycin A1 was added for final 16 h of treatment. **(F)** Citrate synthase activity in HeLa cells after 1 mM DFP treatment for 24 h in cells transfected with siRNA against autophagy genes ATG5 or Beclin1. **(G)** Electron micrographs and **(H)** quantitation from SH-SY5Y cells treated under control conditions or with 1 mM DFP for 16 h each with 50 nM bafilomycin A1 for final 2 h of treatment. Arrows/arrowheads represent mitochondria surrounded by a limiting membrane. Structures labelled by arrows in panel i are shown magnified in ii (autophagosome) and iii (autolysosome). Scale bar represents 0.9 μm. All results are from three independent experiments. All quantitative data are mean ± s.e.m. *$P < 0.05$, **$P < 0.01$, ***$P < 0.001$, NS, not significant. Baf, bafilomycin A1; DFO, deferoxamine; DFP, deferiprone; EBSS, Earle's balanced salt solution; GFP, green fluorescent protein; siRNA, small interfering RNA. Source data for this figure is available on the online supplementary information page.

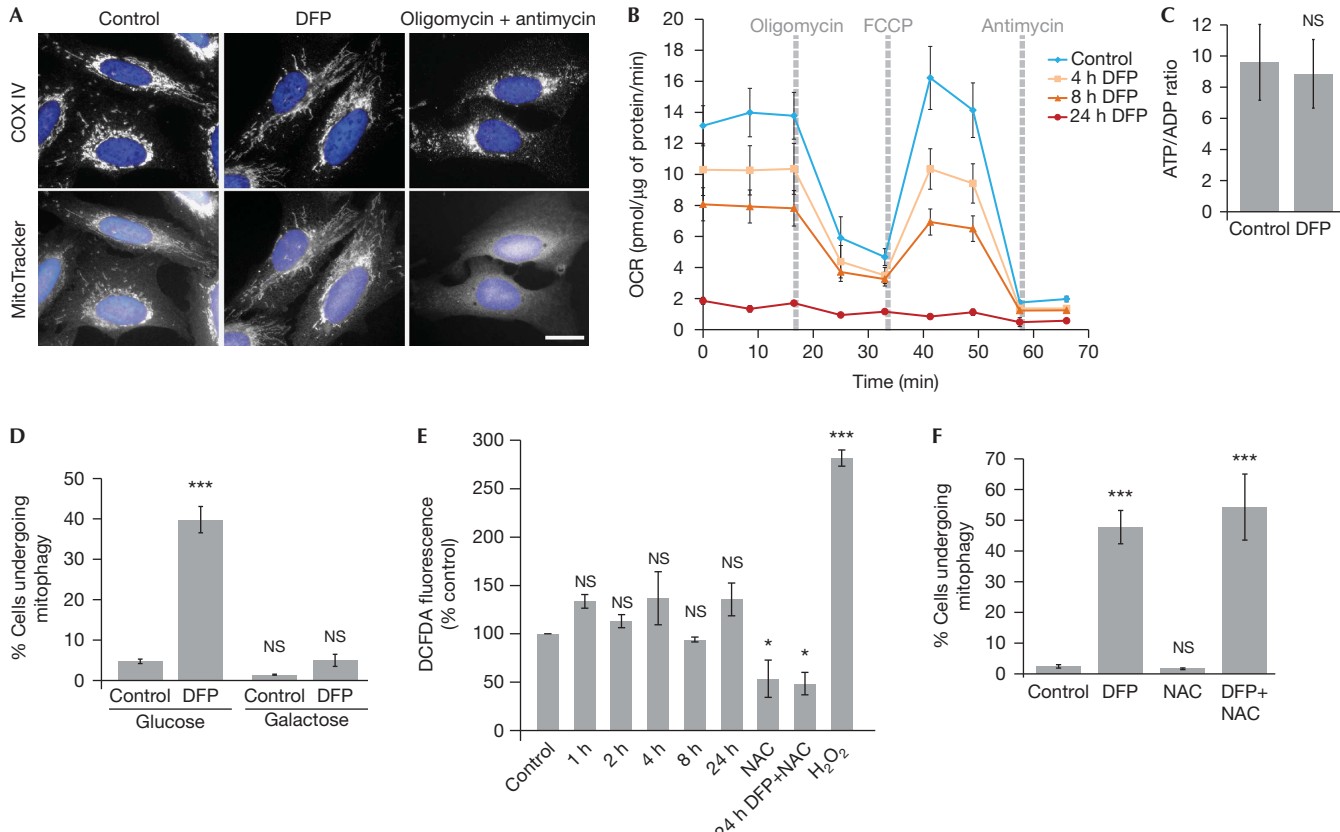

**Fig 3 | Effects of iron chelation on mitochondrial function. (A)** Representative COXIV immunostaining and Mitotracker staining of SH-SY5Y cells treated with 1 mM DFP or 10 μM oligomycin A + 1 μM antimycin A for 24 h. Scale bar, 10 μm. **(B)** Oxygen consumption rate measured after treatment of SH-SY5Y cells with 1 mM DFP for the indicated length of time. 1 μM oligomycin A, 1 μM FCCP and 10 μM antimycin A were injected at the indicated times to determine the proportion of oxygen consumption due to ATP turnover, maximal rate of respiration and amount of proton leak respectively. **(C)** Ratio of ATP to ADP in SH-SY5Y cells treated with 1 mM DFP for 24 h. **(D)** Results of tandem-tag mitophagy assay in SH-SY5Y cells following treatment with 1 mM DFP in cells grown in glucose media or galactose media for 48 h before the addition of DFP. **(E)** Amount of ROS measured by DCFDA fluorescence in SH-SY5Y cells treated with 1 mM DFP for the indicated length of time. 5 mM *N*-acetylcysteine was added for 24 h where indicated. 0.1% $H_2O_2$ was added 5 min before measurement. **(F)** Mitophagy was measured using the tandem tag assay in SH-SY5Y cells treated with 1 mM DFP and/or 5 mM NAC for 24 h. All results are from three independent experiments. All quantitative data are mean ± s.e.m. *$P < 0.05$, ***$P < 0.001$, NS, not significant. DFP, deferiprone; NAC, *N*-acetylcysteine; OCR, oxygen consumption rate; ROS, reactive oxygen species. Source data for this figure is available on the online supplementary information page.

degree of mitophagy that occurs, regardless of the stimulating pathway. It is interesting that respiration loss did not globally affect mitochondrial membrane potential, which might be due to reversal of ATP synthase (Fig 3A).

ROS are required for starvation-induced autophagy [23] and are produced on mitochondrial damage. We only detected a slight increase in ROS production on iron chelation, which was effectively decreased by the scavenger *N*-acetylcysteine (Fig 3E).

**Fig 4** | Mitophagy is independent of PINK1 and Parkin. (**A**) Representative blot and (**B**) quantitation of PINK1 and Parkin protein levels in SH-SY5Y cells following treatment with 1 mM DFP, 20 μM CCCP or 10 μM oligomycin A + 1 μM antimycin A (O + A) for 24 h. (**C**) Representative blot, (**D**) quantitation of MFN2 protein and (**E**) mitophagy assay from SH-SY5Y cells after treatment with 1 mM DFP, 20 μM CCCP or 10 μM oligomycin A + 1 μM antimycin A (O + A) for 24 h in cells transfected with siRNA against PINK1 (**F**). Representative micrographs and (**G**) quantitation of human primary fibroblasts expressing the mCherry-GFP-FIS1$_{101-152}$ construct from a control individual (wild type) and an early-onset Parkinson's disease patient with compound heterozygous Parkin mutations (mutant). Cells were treated with 1 mM DFP, 20 μM CCCP or 10 μM oligomycin A + 1 μM antimycin A (O + A) for 24 h. Scale bar, 10 μm. (**H**) Representative blots of mitochondrial proteins, the autophagy marker LC3 and Parkin and (**I**). Quantitation of HSP60 protein levels in Parkin wild type and mutant human primary fibroblasts treated with 1 mM DFP or 10 μM oligomycin A + 1 μM antimycin A (O + A) for 24 h, 50 nM bafilomycin A1 was added for the final 16 h of treatment. All results are from three independent experiments. All quantitative data are mean ± s.e.m. *$P < 0.05$, **$P < 0.01$, ***$P < 0.001$, NS, not significant. DFP, deferiprone; GFP, green fluorescent protein; MFN2, mitofusin2; siRNA, small interfering RNA. Source data for this figure is available on the online supplementary information page.

$N$-acetylcysteine did not significantly impair DFP-induced mitophagy, implying ROS signalling might differ in mitophagy compared with general autophagy (Fig 3F). This also suggests that mitophagy induced by loss of iron might not be a damage response, but a recycling one. Mitochondria contain large amounts of iron and this form of mitophagy could act to recycle some of this pool to maintain other essential iron-dependent functions. In support, iron chelation can trigger autophagy of ferritin, so autophagy can be a route for iron metabolism [24]. Also, mutation of an autophagy gene was recently found to cause the neurodegenerative and brain iron accumulation disease static encephalopathy with neurodegeneration in adulthood [25].

## Mitophagy is independent of PINK1 and Parkin

The role of PINK1 and Parkin in mitophagy is unclear following findings that endogenous Parkin expression is not sufficient to allow depolarization-induced mitophagy [8]. Our results indicate CCCP and valinomycin, previously shown to activate PINK1 [12], can induce mitophagy in cells where Parkin is not overexpressed (Fig 1C) and it is possible that iron chelation stimulates the same pathway. However, we found that iron chelation still induced mitophagy in HeLa cells (Fig 2F) that reportedly do not express Parkin [26]. To clarify this, we looked at PINK1 stability. PINK1 is imported into mitochondria and cleaved by the protease PARL in the intermembrane space. Upon depolarization, import is prevented and PINK1 is stabilized on the outer mitochondrial membrane and recruits Parkin [27]. Following treatment of cells with CCCP or oligomycin/antimycin combination, the amount of mitochondrial PINK1 was dramatically increased compared with control (Fig 4A,B, supplementary Fig S4B online). In contrast, there was no increase in PINK1 with DFP treatment suggesting the PINK1/Parkin pathway is not activated by iron chelation. We also noted loss of Parkin on CCCP or oligomycin/antimycin treatment (Fig 4A,B), which correlates with the previous observations suggesting Parkin activation leads to autoubiquitylation and increased proteasomal turnover [8]. In support, we find this loss is partially rescued with bortezomib (supplementary Fig S4C online). As with PINK1 stabilization, DFP treatment did not lead to loss of Parkin, further strengthening the notion that the iron chelation pathway is PINK1/Parkin independent (Fig 4A,B).

To investigate the PINK1/Parkin pathway further, we employed siRNA to deplete PINK1 from cells. Loss of PINK1 was sufficient to prevent turnover of MFN2 (which has been shown to be dependent on the PINK1/Parkin pathway in response to mitochondrial depolarization [16]), but did not alter mitophagy in

response to DFP (Fig 4C–E). Surprisingly, loss of PINK1 had no effect on CCCP or oligomycin/antimycin-induced mitophagy suggesting PINK1, and MFN2 turnover, is dispensable for mitophagy. However, the remnant of PINK1 remaining following siRNA could be sufficient to induce mitophagy. Like DFP-induced mitophagy, CCCP or oligomycin/antimycin-induced mitophagy could be blocked by bafilomycin (supplementary Fig S4D online).

As a final confirmation, we obtained primary human dermal fibroblasts from a healthy individual and from a patient suffering from early-onset Parkinson's disease, attributed to mutations in the *PARK2* gene that codes for Parkin (compound heterozygous for a 255delA nucleotide deletion causing a premature truncation and an EXON 3–4 deletion). By expressing our mitophagy marker, we saw significant DFP-induced mitophagy in control fibroblasts and this correlated well with bafilomycin-sensitive loss of mitochondrial markers pyruvate dehydrogenase and HSP60 by western blot (Fig 4F–I). In the Parkinson's fibroblasts that lack full length Parkin, DFP was still able to stimulate mitophagy as efficiently as control cells. In control fibroblasts, the mitophagy tag indicated oligomycin/antimycin and CCCP treatment also increased mitophagy over basal conditions, although this was less than observed in SH-SY5Y cells (Fig 4F,G). As with the neuroblastoma cells, loss of Parkin was observed following oligomycin/antimycin treatment, implying activation of the pathway (Fig 4H). The Parkin mutant cells had a higher basal level of mitophagy compared with control and did not undergo further stimulation with oligomycin/antimycin (Fig 4F,G). In contrast, CCCP resulted in a twofold mitophagy increase, even though there was no detectable Parkin expression. Taken together with the siRNA data, this suggests that under mitochondrial depolarization conditions, the PINK1/Parkin pathway is activated but not required for mitophagy. Although we can detect oligomycin/antimycin/CCCP-induced mitophagy using our fluorescence assay, we were unable to detect flux of mitochondrial proteins by western blot, which highlights the sensitivity of our assay over currently used methods. Regardless, primary fibroblasts, SH-SY5Y and HeLa cells all displayed a comparable and robust induction of mitophagy on iron chelation, independent of the status of the PINK1/Parkin pathway as measured by multiple methods.

DFP and deferoxamine are both clinically available drugs for the treatment of β-thalassaemia and their potential use as anti-neurodegenerative agents has been the subject of much debate [28]. The fact that iron accumulation is common to a range of neurodegenerative diseases including Friedreich's ataxia, Parkinson's and Alzheimer's disease, raises the possibility that iron levels might be critical in determining mitophagy: although we have shown here that loss of iron triggers mitophagy, it is

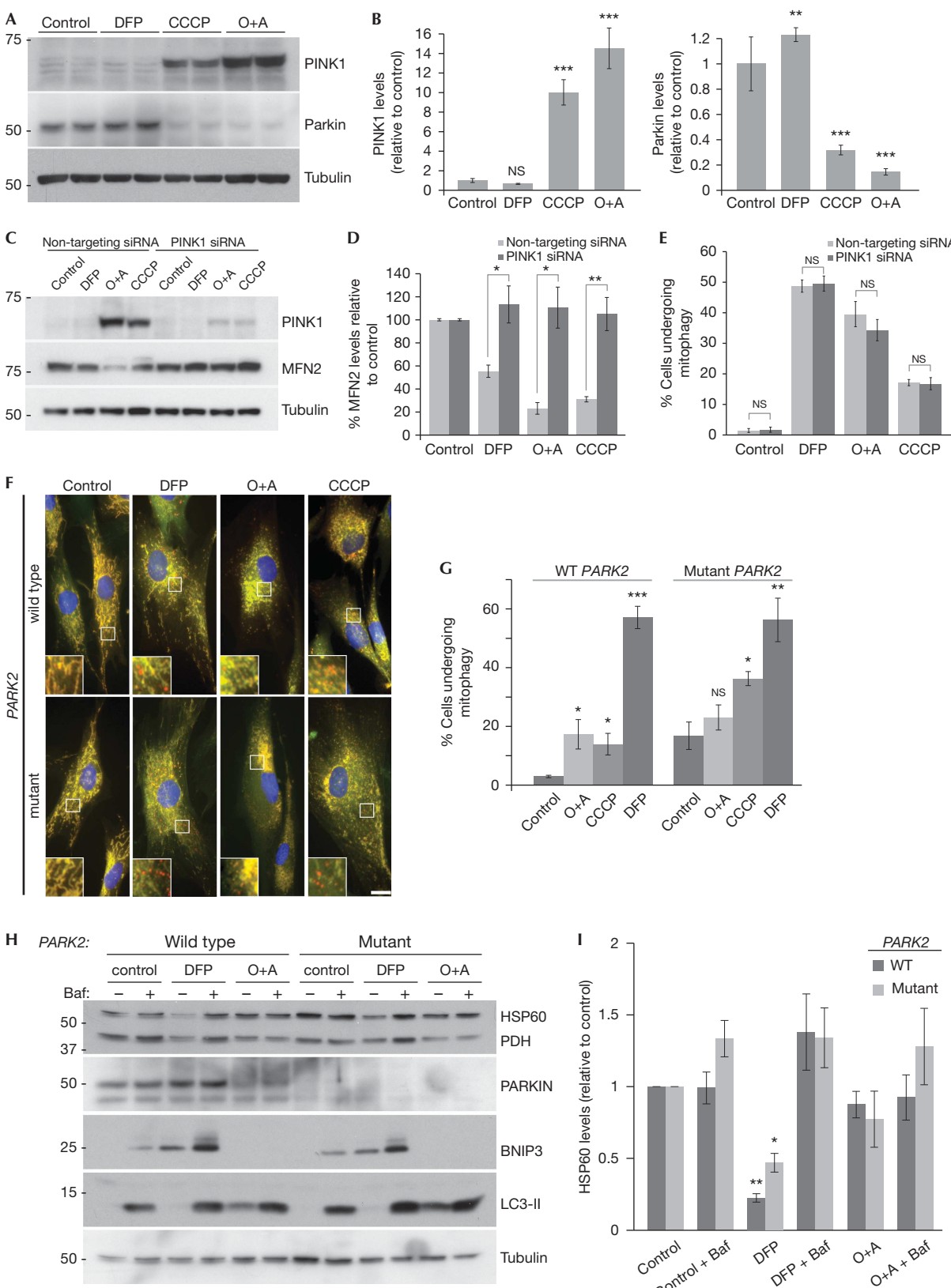

possible that accumulation of iron might prevent mitophagy. We are currently investigating whether this is the case, but acknowledge that given incubation in galactose media severely impairs mitophagy, iron chelation might not be as relevant in cells that are reliant on oxidative phosphorylation such as neurons. Regardless, our results demonstrate a hitherto unknown effect of iron levels on mitophagy. Further work to determine the molecular signalling underlying mitophagy induced by loss of iron will hopefully lead to the discovery of more specific activators of this pathway and potential therapeutic leads for multiple diseases.

## METHODS

**Cell culture and treatment.** Human primary skin fibroblasts Control ND34791 and Parkin mutant ND29543 were purchased from the Coriell Institute (Camden, NJ, USA) and all other cell lines were from ATCC (Teddingon, UK). Fibroblasts, U2OS and HeLa cells were cultured at $37\,°C/5\%$ $CO_2$ and maintained in Dulbecco's Modified Eagle Medium (DMEM) media with 10% fetal bovine serum and antibiotics. SH-SY5Y cells were grown in DMEM/F-12 media with 15% fetal bovine serum and antibiotics.

**DNA construct and expression.** cDNA for mCherry, GFP and residues 101–152 of human FIS1 were cloned into a pBABE.hygro vector. The construct was co-transfected into 293FT cells with GAG/POL and VSV-G expression plasmids (Clontech, Saint-Germain-en-Laye, France) for retrovirus production using Lipofectamine 2000 (Life Technologies) in accordance with manufacturer's instructions. Virus was harvested 48 h after transfection and applied to cells in the presence of 10 µg/ml polybrene. Cells were selected with 500 µg/ml hygromycin and stable pool used for experiments.

**Mitophagy assay.** Cells stably expressing mCherry-GFP-FIS1$_{101-152}$ were plated on glass coverslips and treated as described. Coverslips were washed twice with phosphate-buffered saline, fixed with 3.7% formaldehyde, 50 mM HEPES pH7.0 for 10 min, washed twice with and then incubated for 10 min with DMEM, 10 mM HEPES pH7.4, washed once with phosphate-buffered saline and mounted using Prolong gold mounting solution with 4,6-diamidino-2-phenylindole (Life Technologies) and visualized using a Nikon Ti-S fluorescence microscope. Images were quantified by manual counting using NIS-Elements (Nikon) software and images processed using Adobe Photoshop.

**Mitophagy assay quantitation.** For all conditions tested, quantitation was performed on at least three fields of view ($>48$ cells per condition per experiment) except for primary fibroblasts that because of the large size of the cells at least 10 fields of view were quantified ($>17$ cells per condition per experiment). Red-alone puncta were defined as round structures found only in the red channel with no corresponding structure in the green channel. Quantitative data were collected by counting all of the red-alone puncta within each cell for each field of view. Size of individual puncta was not considered during data collection, although it should be noted that puncta were relatively uniform in size. Intensity of puncta was not considered although all counted red-alone puncta were visible on the merged image as well as in the red channel alone. Following data collection, a threshold of 3 or more red-alone puncta per cell was applied to the data to determine the number of cells undergoing mitophagy. Data were not collected with the counter blinded to condition except where stated.

**Statistical analysis.** Statistical significance was determined using unpaired Student's $t$-test for single comparisons (Figs 2D,F,H, 3C and 4D,E,G, supplementary Fig S2A online) and for multiple treatments, analysis of variance was performed followed by Fisher's Least Significant Difference test for individual comparisons (Figs 1C,E,G, 2B,E, 3D,E,F and 4B,I, supplementary Figs S1D–F, S3A,B,D and S4D online). In all cases, individual comparisons were made to the control condition and $P<0.05$ was considered significant. Data were analysed using GraphPad Prism 6 software (GraphPad Software La Jolla, CA, USA).

ACKNOWLEDGEMENTS
We thank the Ganley lab for discussions and manuscript reading. Thanks to the support of the MRC Protein Phosphorylation and Ubiquitylation Unit DNA Sequencing Service (coordinated by Nicholas Helps) and the tissue culture team (coordinated by Kirsten Airey). This work was supported by the Medical Research Council, a Wellcome Trust Strategic Award (097945/B/11/Z) and by the companies supporting the Division of Signal Transduction Therapy Unit (AstraZeneca, Boehringer-Ingelheim, GlaxoSmithKline, Merck KGaA, Janssen Pharmaceutica and Pfizer).

  *Author contributions*: G.F.G.A. and I.G.G. conceived/designed/performed the experiments, analysed data and wrote the paper. R.T. performed molecular biology and J.J. electron microscopy.

CONFLICT OF INTEREST
The authors declare that they have no conflict of interest.

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
