## [Review Process File · EMBO Reports]

Manuscript EMBOR-2013-37610

Loss of iron triggers PINK1/Parkin-independent mitophagy

George F.G. Allen, Rachel Toth, John James and Ian G. Ganley

Corresponding author: Ian G. Ganley, University of Dundee

Review timeline:

Submission date:	04 June 2013
Editorial Decision:	04 July 2013
Revision received:	09 September 2013
Editorial Decision:	27 September 2013
Revision received:	30 September 2013
Accepted:	01 October 2013

Transaction Report:

Editor: Nonia Pariente

1st Editorial Decision

04 July 2013

Thank you for your submission to EMBO reports. We have now received the reports of the three referees that were asked to evaluate your study, which can be found at the end of this email. As you will see, although all the referees find the topic of interest and in principle suitable for us, they also raise several issues that need to be experimentally addressed before publication can be considered.

Given that all referees are in fair agreement and provide constructive suggestions on how to strengthen the work, I would like to give you the opportunity to revise your manuscript. The most important concerns to be addressed are the inclusion of thorough statistical analyses throughout the study (please note that they need to be performed on a minimum of three independent experiments), to show that the method you describe can indeed monitor parkin-dependent mitophagy under endogenous parkin conditions, which is a main claim of the work, improve the characterization of iron chelation as indicated by referee 2 and of specific mitophagy induction (as opposed to general macroautophagy). In addition, please also address all minor issues raised by the referees. Addressing referee 3's point 8, however, although clearly of interest, would not be a precondition for publication.

If the referee concerns can be adequately addressed, we would be happy to consider your manuscript for publication. However, please note that it is EMBO reports policy to undergo one round of revision only and thus, acceptance of your study will depend on the outcome of the next, final round

of peer-review.

Revised manuscripts must be submitted within three months of a request for revision unless previously discussed with the editor; they will otherwise be treated as new submissions. Revised manuscript length must be a maximum of 28,500 characters (including spaces). When submitting your revised manuscript, please also include editable TIFF or EPS-formatted figure files, a separate PDF file of any Supplementary information (in its final format) and a letter detailing your responses to the referees.

We now encourage the publication of original source data -particularly for electrophoretic gels and blots- with the aim of making primary data more accessible and transparent to the reader. If you agree, you would need to provide one PDF file per figure that contains the original, uncropped and unprocessed scans of all or key gels used in the figures. The PDF files should be labeled with the appropriate figure/panel number, and should have molecular weight markers; further annotation could be useful but is not essential. The PDF files will be published online with the article as supplementary "Source Data" files and should be uploaded when you submit your final version. If you have any questions regarding this please contact me.

We also welcome the submission of cover suggestions or motifs that might be used by our Graphics Illustrator in designing a cover.

I look forward to seeing a revised form of your manuscript when it is ready. In the meantime, do not hesitate to get in touch with me if I can be of any assistance.

REFEREE REPORTS:

Referee #1:

This is a very interesting and well-written manuscript describing a novel cell-based assay of mitophagy, and a novel mitophagy-inducing pathway. The experiments validating the mitophagy assay are extremely rigorous, and their assay will be useful to many investigators studying mitophagy. While the author's do not describe the mechanism underlying the novel mitophagy pathway revealed from their work, they do show that this pathway is induced upon iron chelation, that it involves a metabolic switch from oxidative phosphorylation to glycolysis, and that it is independent of the PINK1-Parkin pathway. While I am very enthusiastic about this manuscript, there are some issues the author's need to address, which I have summarized below.

-The authors make many comparisons between treatments, but their paper is completely devoid of statistical analysis. While in some cases it is possible to infer that differences are significant, in many cases, this is not evident (e.g., Figure 1C). It would also be nice to see the p values for many of the comparisons that they make. In short, appropriate statistical analyses of all of the data involving comparisons must be included in the manuscript.

-The authors do not indicate whether cells were scored blinded to condition. This is important, and hopefully how the data was collected. The authors need to address this matter.

-The authors argue that their assay is sensitive enough to detect Parkin-dependent mitophagy under endogenous (non-Parkin overexpressing) conditions. Since there is no data to show that the mitophagy they see is Parkin-dependent, the author's should either strike this claim from their manuscript, or provide the appropriate data to back up this claim.

-The authors need to expand their Supplemental Methods section on immunoblotting. It mentions nothing about the quantification of the blots-what software they used, whether they normalized their WB signal to the loading control, etc. This applies to Figs. 2d, 4d, 4f, and S2d.

-There are three parts of the paper (Figs. 2f, 4a-b, and 4e-f) in which quantification was performed

based on only 2 independent experiments. The authors should bring the N up to 3 if they are going to show standard error bars and perform statistical analyses.

-The authors are unclear about how the EMs were scored in the legend to Figure 2. They say that "structures containing mitochondria were counted in 3 cell sections/treatment." What is a "cell section"? Is it a cut through the entire block of fixed cells? Is it a portion of the block's horizontal area? How many cells does it contain?

-In the text referring to Fig. 2c, the authors say that p62 "displayed flux." It is unclear what they mean by this. There is an increase in p62 abundance in DFP-treated cells, but nowhere near as much as the abundance increase after treatment with both DFP and bafilomycin. Also, the p62 abundance was higher in DFP/bafilomycin cells than in vehicle/bafilomycin cells. Taken together, these findings indicate both induction and unimpaired autophagic turnover of p62. This interpretation does not seem to fit my definition of the term "flux." The author's need to clarify this matter.

-On p. 7, the authors write, "In both cases, we could not detect a change in flux at the protein level, highlighting the sensitivity of our assay." They need to be clear about what "both cases" refers to. Do they mean the mutant and WT O&A lanes?

-On p. 7 the authors say that baseline mitophagy is "slightly" higher in parkin fibroblasts, but the increase looks like it is at least 3-fold, which in my book is a lot more than slight.

-On p. 4 the authors refer to Fig S3 when they really mean Fig S2.

-The author's cite Ziviani, et al in support of the notion that mitofusins are degraded to prevent re-fusion of lysosome-destined mitochondria. Richard Youle's lab performed an experiment to address this matter in published work (Tanaka, et al 2010 JCB 191, 1367), so it would be more appropriate to cite Richard's paper.

-Finally, the authors claim that activation of their novel mitophagy pathway could prove therapeutic in diseases such as Parkinson's disease. Given that their mitophagy pathway cannot be activated in cells dependent on respiration (like neurons), I am skeptical about the therapeutic potential of this strategy. I realize that these sorts of claims help sell papers, but they also create unrealistic expectations. There are many interesting findings in this manuscript that are worthy of publication, so I would encourage the authors to tone down their claims of therapeutic potential; they really aren't necessary to sell the paper in this case.

Referee #2:

This work from Allen and colleagues describes the establishment of a novel assay for mitophagy, its use in a chemical screen for modulators of mitophagy and the further characterization of an iron chelator, DFP, as a potent inducer of mitophagy. Overall, this is a nice study that extends our insight into the process and regulation of mitophagy, which could ultimately be useful in therapeutic intervention of neurodegenerative diseases such as Parkinson's. However, I have the following comments and recommendations:

The central assay is neat and appears robust, based on the similar use of RFP-GFP-LC3, and appears to confer the desired readout. However, it would be useful to see a better description about the scoring system in the main text or in Methods since I had to go searching for this buried in the figure legend. This is obviously of critical importance to the validity of the assay and so is worthy of a little more description, e.g. if the cells were scored manually for >3 red puncta was there any consideration for puncta size, relative redness etc. Since it isn't stated one assumes this wasn't done with the scorer blinded to conditions. This is also worth noting clearly. I would consider it appropriate to show that the 'red only' puncta do actually co-localise with lysosome (LAMP1 staining) as suspected.

The characterization that iron chelation is responsible for the observed effects is not thoroughly

covered. Although the majority of the work focuses on DFP, this is complemented by only one set of experiments analyzing alternative iron chelators DFO and Dp44mT. Have the authors tested how well these chelate the available iron? Can the authors quantify free/bound iron in the presence or absence of these chelators? Does their iron chelating characteristics correlate with their ability to induce mitophagy in this assay? What about other classic iron chelators? Further, it would be nice to see a dose response, at least for DFP.

It should at least be discussed what other things (metal ions, cellular processes etc) might be affected by DFP and others.

I do not agree with the comment on p4 "showing that iron chelation is specifically inducing mitochondrial turnover rather than general autophagy". A general autophagy inducer (starvation, TOR inhibition) surely would also lead to the increase in COXIV-LC3 co-localising puncta in a non-specific manner, but perhaps in comparison DFP induces a greater proportion of co-localising puncta. In any case, this would be a relevant comparison to make given that a main claim is that the effect is specifically inducing mitophagy. I appreciate that a TOR inhibitor (Ku-00063794) is described here but a more 'typically' one (rapamycin/Torin1) should be used to compare (see comment below).

The nature of the Parkinson's patient's mutation is not clear. Is 255delA nucleotide? 255A doesn't correspond to protein sequence.

It is striking that not a single statistical test has been done. Have the authors verified that there effects are so robust that this is not necessary?

Minor:

p4. "Additionally, two other structurally distinct iron chelators (deferoxamine and Dp44mT) stimulated mitophagy (Fig. S3)". This should refer to Fig. S2.

Also, one presumes deferoxamine is what is abbreviated as DFO. This should be stated in the text.

Fig. 2 legend. I don't understand what the "[39][39][38][41]" and "[38][38][37][37]" are. It doesn't fit with the references (especially as there isn't a ref 41). Maybe these are typos?

Fig. S2. CoCl₂ and DMOG are not discussed. Some reference should be made to the rationale for their inclusion in the results or should be removed.

Similarly, I am not familiar with the efficacy of Ku-00063794 which is not cited nor the source given.

The cell culture conditions for growth under galactose vs glucose medium was not clear.

Referee #3:

This is an interesting report by Allen and colleagues who have developed a dual-fluorescence-based assay to track mitochondrial engulfment by the lysosome. Importantly, they have used it to demonstrate that iron chelation promotes mitophagy in a parkin/PINK1-independent manner. Although the concept of the tandem fluorescence autophagy reporter is not new, it is rather clever of the authors to apply this technique to assay for mitophagy. Overall, the main finding is novel notwithstanding the lack of mechanistic insights on how the chelation of iron results in mitochondrial clearance.

I do have a number of comments/suggestions that might help the authors to improve the manuscript further, as listed below.

1. A consistent problem with the quantitative data provided in this paper is the lack of statistical evaluation. Please provide statistics for all the bar-graphs presented.
2. In characterizing the utility of the mCherry-GFP-Fis1 reporter, it will be informative to

supplement the time course study shown in Fig. 1F with images showing the localization of the reporter in cells treated with DFP to examine the progressive changes that take place with time. Although the authors stated that DFP-mediated mitophagy does not involve global changes in mitochondrial localization, they nonetheless noted that those undergoing mitophagy were separated from the network.

3. The authors would need to clarify the status of endogenous parkin in U2OS osteosarcoma cells before stating that their "assay is sensitive enough to observe parkin-dependent mitophagy under endogenous parkin conditions (p.3)". (Many cancer cell lines are deficient in parkin expression). Importantly, are the DFP-induced mitochondrial puncta positive for parkin or PINK1 in these cases?

4. Fig. 2D: Control (instead of control + Baf) would be more useful here to illustrate the depletion of mitochondrial-localized proteins in the presence of DFP and their impaired degradation in the presence of DFP + Baf. Further, it would be beneficial to include an additional lane to show the effect of proteasome inhibition on the level of the proteins examined (especially Mfn2) to support the author's suggestion that Mfn2 is degraded by the proteasome in a DFP-dependent manner. This is important to clarify as Mfn2 degradation is thought to be initiated by parkin-mediated ubiquitination, which is apparently not essential here.

5. Curiously, the level of p62 appears to increase in DFP treated cells relative to control alone (Fig. 2D). This begs the question on whether DFP-induced mitophagy requires p62, which plays a contentious role in the parkin/PINK1 mitophagy pathway. Related to this, is K63-linked ubiquitination involved here, as parkin is thought to perform this mode of ubiquitination in preparing the damaged organelle for clearance by mitophagy?

6. It is curious to note in Fig. 4A that despite the abundance of full length PINK1 (which serves to recruit parkin) in cells treated with either CCCP or oligomycin/antimycin, the expression of parkin is significantly reduced. The authors suggested that parkin in this case is degraded by the proteasome, which is easily testable using proteasome inhibitor treatment.

7. Show anti-parkin staining in Fig. 4C.

8. Considering the current debate surrounding the role of parkin/PINK1 pathway in neurons, it would be interesting to examine if DFP induces mitophagy in primary neurons using the author's tandem reporter assay (particularly in view of the author's proposal that iron accumulation in AD or PD brains may compromise mitophagy).

1st Revision - authors' response

09 September 2013

Referee #1:

This is a very interesting and well-written manuscript describing a novel cell-based assay of mitophagy, and a novel mitophagy-inducing pathway. The experiments validating the mitophagy assay are extremely rigorous, and their assay will be useful to many investigators studying mitophagy. While the author's do not describe the mechanism underlying the novel mitophagy pathway revealed from their work, they do show that this pathway is induced upon iron chelation, that it involves a metabolic switch from oxidative phosphorylation to glycolysis, and that it is independent of the PINK1-Parkin pathway. While I am very enthusiastic about this manuscript, there are some issues the author's need to address, which I have summarized below.

We thank the reviewer for their time, kind comments and constructive suggestions.

1. The authors make many comparisons between treatments, but their paper is completely devoid of statistical analysis. While in some cases it is possible to infer that differences are significant, in many cases, this is not evident (e.g., Figure 1C). It would also be nice to see the p values for many

of the comparisons that they make. In short, appropriate statistical analyses of all of the data involving comparisons must be included in the manuscript.

We apologise for the absence of statistical analysis in the original submission. All data has now been analysed, using either Student's T-test or ANOVA followed by Fisher's Least Significant Difference test. Where a difference in observation was inferred in the original submission, statistical analysis has been carried out to show significance with a p-value of 0.05 or lower. This analysis is included in all figures and a description of the tests are given in the Supplemental Methods section.

From Supplementary Methods:

“Statistical Analysis

Statistical significance was determined using unpaired Student's t-test for single comparisons (Figures 2D, 2F, 2H, 3C, 4D, 4E, 4G, S2A) and for multiple treatments ANOVA was performed followed by Fisher's Least Significant Difference Test for individual comparisons (Figures 1C, 1E, 1G, 2B, 2E, 3D, 3E, 3F, 4B, 4I, S1D, S1E, S1F, S3A, S3B, S3D, S4D). In all cases individual comparisons were made to the control condition and $P < 0.05$ was considered significant. Data was analysed using GraphPad Prism 6 software (GraphPad Software Inc. La Jolla, CA, USA).”

2. The authors do not indicate whether cells were scored blinded to condition. This is important, and hopefully how the data was collected. The authors need to address this matter.

The screen for mitophagy inducers shown in Fig. 1C was scored blinded and this is now noted in the figure 1 legend. Other experiments were not scored blind and this is also now noted in the supplementary methods. We feel that our method of counting is unbiased despite for the most part not being counted blind. All red-alone puncta were counted in all cells within each field of view and therefore we feel this reduces the likelihood of bias being introduced. We have also improved our description of our counting method in supplementary methods.

From Figure 1 legend:

“C. Screen for mitophagy inducing conditions using the tandem tag mitophagy assay in SHSY5Y cells. All treatments 24h. For this experiment quantitation of mitophagy was performed with counter blinded to condition.”

From Supplementary methods:

“Mitophagy Assay Quantitation

For all conditions tested quantitation was performed on at least three fields of view (>48 cells per condition per experiment) except for primary fibroblasts that due to the large size of the cells at least 10 fields of view were quantified (>17 cells per condition per experiment). Red alone puncta were defined as round structures found only in the red channel with no corresponding structure in the green channel. Quantitative data was collected by counting all of the red alone puncta within each cell for each field of view. Size of individual puncta was not considered during data collection although it should be noted that puncta were relatively uniform in size. Intensity of puncta was not considered although all counted red-alone puncta were visible on the merged image as well as in the red channel alone. Following data collection a threshold of 3 or more red alone puncta per cell was applied to the data to determine the number of cells undergoing mitophagy. Data was not collected with the counter blinded to condition except where stated.”

3. The authors argue that their assay is sensitive enough to detect Parkin-dependent mitophagy under endogenous (non-Parkin overexpressing) conditions. Since there is no data to show that the

mitophagy they see is Parkin-dependent, the author's should either strike this claim from their manuscript, or provide the appropriate data to back up this claim.

We agree with the reviewer that we had not conclusively demonstrated a role for the Parkin pathway in depolarization-induced mitophagy. To test this we carried out further experiments shown in Fig. 4 and copied below for convenience. Firstly, we carried out PINK1 siRNA in SH-SY5Y neuroblastoma cells and found that even though depletion was at a sufficient level to impair turnover of MFN2 (previously shown to depend on PINK1-Parkin function), its loss did not alter mitophagy. We were surprised by this result and therefore carried out further experiments in our primary human Parkin-deficient cells. Initially we had induced mitophagy using igomycin/antimycin combination and have expanded our study to use CCCP. CCCP induced a two-fold increase in mitophagy over basal levels in both control and Parkin cells (and in the control cells we observed a loss in Parkin levels implying activation of Parkin as in the SH-SY5Y cells). We do not know why CCCP-induced mitophagy is higher in the mutant Parkin cells than in WT, but this could be due to the high baseline in the mutants or heterogeneity often observed in primary human samples. Regardless, the data from the neuroblastoma and primary cells suggests the Parkin pathway is not essential for mitophagy under these conditions. We have modified the text to remove mention of our assay being able to detect endogenous Parkin-dependent mitophagy and have added the following conclusion.

From Results and Discussion, Mitophagy is independent of PINK1 and Parkin:

“Taken together with the siRNA data, this suggests that under mitochondrial depolarisation conditions, the PINK1/Parkin pathway is activated but not required for mitophagy.”

4. The authors need to expand their Supplemental Methods section on immunoblotting. It mentions nothing about the quantification of the blots-what software they used, whether they normalized their WB signal to the loading control, etc. This applies to Figs. 2d, 4d, 4f, and S2d.

We apologise for the omission. Western blots were normalized to tubulin (as a loading control) and quantified using ImageJ. This information has been added to the Supplemental Methods section.

From Supplementary Methods, Immunoblotting:

"X-ray film was scanned using an Epson perfection V700 photo scanner with Epson Scan 3.28E software (Epson, Hemel Hempstead, UK). Western blotting data was quantified using Image J software (<http://rsbweb.nih.gov>) and all data was normalised to the loading control α -tubulin."

5. There are three parts of the paper (Figs. 2f, 4a-b, and 4e-f) in which quantification was performed based on only 2 independent experiments. The authors should bring the N up to 3 if they are going to show standard error bars and perform statistical analyses.

All experiments have now been carried out a minimum of three times. The representative blots shown for Figs. 2F, 4A-B and 4E-F (now H-I) remain the same as the original submission, but the new data has been incorporated into the quantitation graphs. Statistical analysis has been performed on the quantitative data as described for point 1.

6. The authors are unclear about how the EMs were scored in the legend to Figure 2. They say that "structures containing mitochondria were counted in 3 cell sections/treatment." What is a "cell section"? Is it a cut through the entire block of fixed cells? Is it a portion of the block's horizontal area? How many cells does it contain?

We apologise for the ambiguous description. We have changed the axis title to clarify our quantitation "Autophagosomal structures containing mitochondria per 50 cell sections". We counted 50 cells (not whole cells but sections of cells on each thin resin slice) per experiment and carried out three independent experiments. This is now explained in the Supplemental Methods section.

From Supplementary Methods, Electron Microscopy:

"Images were quantified by manual counting of single or double membrane structures containing mitochondria. 50 cell sections (cross sections through 50 cells) were counted per condition for each independent experiment."

7. In the text referring to Fig. 2c, the authors say that p62 "displayed flux." It is unclear what they mean by this. There is an increase in p62 abundance in DFP-treated cells, but nowhere near as much as the abundance increase after treatment with both DFP and bafilomycin. Also, the p62 abundance was higher in DFP/bafilomycin cells than in vehicle/bafilomycin cells. Taken together, these findings indicate both induction and unimpaired autophagic turnover of p62. This interpretation does not seem to fit my definition of the term "flux." The author's need to clarify this matter.

We understand the reviewers point on the use of the term flux. We used the term flux to indicate that the turnover of p62 was not blocked as it could be increased by bafilomycin treatment. We apologise for using this term ambiguously. We have modified the text as shown below.

From Results and Discussion, Loss of iron induces mitophagy:

"Expression of the autophagy adaptor and substrate SQSTM1/p62 increased following mitophagy induction and the amount of p62 was further enhanced with bafilomycin treatment. This suggests a role for p62 in DFP-induced mitophagy, though work is needed to clarify this."

8. On p. 7, the authors write, "In both cases, we could not detect a change in flux at the protein level, highlighting the sensitivity of our assay." They need to be clear about what "both cases" refers to. Do they mean the mutant and WT O&A lanes?

This is what we meant and we apologise for the confusion and have modified the text to remove this confusing statement.

From Results and Discussion, Mitophagy is independent of PINK1 and Parkin:

"Though we can detect oligomycin/antimycin/CCCP-induced mitophagy using our fluorescence assay, we were unable to detect flux of mitochondrial proteins by western blot, which highlights the sensitivity of our assay over currently used methods."

9. On p. 7 the authors say that baseline mitophagy is "slightly" higher in parkin fibroblasts, but the increase looks like it is at least 3-fold, which in my book is a lot more than slight.

We apologise for not being clear, we understand the reviewers point and have modified the text.

From Results and Discussion, Mitophagy is independent of PINK1 and Parkin:

"The Parkin mutant cells had a higher basal level of mitophagy compared to control and did not undergo further stimulation with oligomycin/antimycin (Fig. 4F-G)."

10. On p. 4 the authors refer to Fig S3 when they really mean Fig S2.

This has been corrected, though the supplemental figures have changed.

11. The author's cite Ziviani, et al in support of the notion that mitofusins are degraded to prevent re-fusion of lysosome-destined mitochondria. Richard Youle's lab performed an experiment to address this matter in published work (Tanaka, et al 2010 JCB 191, 1367), so it would be more appropriate to cite Richard's paper.

We apologise for mis-referencing the work and now cite Tanaka, et al. (*Proteasome and p97 mediate mitophagy and degradation of mitofusins induced by Parkin*. The Journal of cell biology, 2010. **191**(7): 1367-80) as reference 16.

12. Finally, the authors claim that activation of their novel mitophagy pathway could prove therapeutic in diseases such as Parkinson's disease. Given that their mitophagy pathway cannot be activated in cells dependent on respiration (like neurons), I am skeptical about the therapeutic potential of this strategy. I realize that these sorts of claims help sell papers, but they also create unrealistic expectations. There are many interesting findings in this manuscript that are worthy of publication, so I would encourage the authors to tone down their claims of therapeutic potential; they really aren't necessary to sell the paper in this case.

We appreciate the reviewers concerns and are thankful for the comments on the worthiness of the data. We have tried to tone down the use of mitophagy in neurons as a therapeutic approach, but we do genuinely hope some therapeutic benefit will come from these studies eventually, potentially in other tissues or perhaps other cell types in the brain where the reliance on oxidative phosphorylation is not as high. We have modified the text in the concluding paragraph as shown below.

From Results and Discussion, Mitophagy is independent of PINK1 and Parkin:

“DFO and DFP are both clinically available drugs for the treatment of β -thalassemia and their potential use as anti-neurodegenerative agents has been the subject of much debate [28]. The fact that iron accumulation is common to a range of neurodegenerative diseases including Friedreich’s ataxia, Parkinson’s and Alzheimer’s disease, raises the possibility that iron levels may be critical in determining mitophagy: while we have shown here that loss of iron triggers mitophagy, it is possible that accumulation of iron may prevent mitophagy. We are currently investigating whether this is the case, but acknowledge that given incubation in galactose media severely impairs mitophagy, iron chelation may not be as relevant in cells that are reliant on oxidative phosphorylation such as neurons. Regardless, our results demonstrate a hitherto unknown effect of iron levels on mitophagy. Further work to determine the molecular signalling underlying mitophagy induced by loss of iron will hopefully lead to the discovery of more specific activators of this pathway and potential therapeutic leads for multiple diseases.”

Referee #2:

This work from Allen and colleagues describes the establishment of a novel assay for mitophagy, its use in a chemical screen for modulators of mitophagy and the further characterization of an iron chelator, DFP, as a potent inducer of mitophagy. Overall, this is a nice study that extends our insight into the process and regulation of mitophagy, which could ultimately be useful in therapeutic intervention of neurodegenerative diseases such as Parkinson's. However, I have the following comments and recommendations:

We thank the reviewer for their time and effort and appreciate their constructive comments.

1. The central assay is neat and appears robust, based on the similar use of RFP-GFP-LC3, and appears to confer the desired readout. However, it would be useful to see a better description about the scoring system in the main text or in Methods since I had to go searching for this buried in the figure legend. This is obviously of critical importance to the validity of the assay and so is worthy of a little more description, e.g. if the cells were scored manually for >3 red puncta was there any consideration for puncta size, relative redness etc. Since it isn't stated one assumes this wasn't done with the scorer blinded to conditions. This is also worth noting clearly.

We apologise for the rather brief description of the mitophagy assay in the original submission. We have now added more details in the Supplementary Methods section (shown below for convenience). Also, the original panel of compounds used to screen for autophagy inducers was scored blinded and the text now states this. The other experiments were not performed blind and this is also now stated in supplementary methods.

From Supplementary Methods:

“Mitophagy Assay Quantitation

For all conditions tested quantitation was performed on at least three fields of view (>48 cells per condition per experiment) except for primary fibroblasts that due to the large size of the cells at least 10 fields of view were quantified (>17 cells per condition per experiment). Red alone puncta were defined as round structures found only in the red channel with no corresponding structure in the green channel. Quantitative data was collected by counting all of the red alone puncta within each cell for each field of view. Size of individual puncta was not considered during data collection although it should be noted that puncta were relatively uniform in size. Intensity of puncta was not considered although all counted red-alone puncta were visible on the merged image as well as in the red channel alone. Following data collection a threshold of 3 or more red alone puncta per cell was applied to the data to determine the number of cells undergoing mitophagy. Data was not collected with the counter blinded to condition except where stated.”

We also include example images of the mitophagy induced by the different compounds in Fig. S1C.

2. I would consider it appropriate to show that the 'red only' puncta do actually co-localise with lysosome (LAMP1 staining) as suspected.

We have co-stained cells with LysoTracker stain and the red-only (not the red-green) dots do indeed co-localize with lysotracker. This data is shown in Figure S1B and mentioned in the text (copied below). A limitation of the tandem-tag assay is that when cells are permeabilized for antibody staining (e.g. LAMP1), the pH gradient in the lysosome is disrupted and the GFP signal is restored and therefore red-alone puncta are lost. Thus we were unable to co-stain for LAMP1.

From Results and Discussion, A chemical screen for mitophagy inducers:

“Treatment with several of these compounds for 24h led to mitophagy as indicated by redalone puncta, corresponding to the mitochondrial tag in acidic lysosomes, confirmed by LysoTracker staining (Fig. S1B-C).”

3. The characterization that iron chelation is responsible for the observed effects is not thoroughly covered. Although the majority of the work focuses on DFP, this is complemented by only one set of experiments analyzing alternative iron chelators DFO and Dp44mT. Have the authors tested how well these chelate the available iron? Can the authors quantify free/bound iron in the presence or absence of these chelators? Does their iron chelating characteristics correlate with their ability to induce mitophagy in this assay? What about other classic iron chelators? Further, it would be nice to see a dose response, at least for DFP. It should at least be discussed what other things (metal ions, cellular processes etc) might be affected by DFP and others.

These are important suggestions and we have tried to address them.

We expanded our panel of different iron chelators to five and now show that mitophagy is induced by them all. This data is shown in Fig. S1D.

D.

To correlate iron chelation level and mitophagy, we firstly tried to quantitate free iron using a commercially available kit (Abcam Iron Assay Kit (ab83366)), but were not able to get our samples concentrated enough to get a reading. We therefore looked at a known iron-dependent process within the cell: expression of the transferrin receptor. Depletion of iron results in increased transferrin receptor mRNA stabilization and protein expression through the binding of iron regulatory proteins (Mullner and Kuhn, 1988, Cell 53: 815-825). Using increasing DFP concentrations we were able to increase transferrin receptor levels and this correlated very nicely with the level of mitophagy, implying the mitophagy response is indeed through iron loss. This data is shown in a new panel of Figure 1G and described in the text as shown below.

From Results and Discussion, A chemical screen for mitophagy inducers:

“To directly implicate iron, we found that DFP pre-treatment with Fe³⁺ blocked its ability to stimulate mitophagy (Fig. 1E). Five structurally distinct iron chelators, including deferoxamine (DFO), also stimulated mitophagy (Fig. 1E-G, S1D). To strengthen the link between iron chelators, iron levels and mitophagy, we looked at transferrin receptor levels, which is increased upon intracellular iron depletion [13]. We found a close correlation between mitophagy and transferrin receptor levels in response to iron chelator dose or type (Fig. 1F-G), supporting a role for iron loss in mitophagy. Additionally, the degree of mitophagy peaked at ~24h of treatment with 1mM DFP (Fig. S1E-F).”

We now have a dose response in Fig. 1G and a more detailed one in Fig S1E.

We mention in the text that iron is involved in many cellular functions:

From Results and Discussion, Effects of iron chelation on mitochondrial function:

“We reasoned as the autophagy is specific for mitochondria, iron loss may impair

mitochondrial function that in turn signals for mitophagy. Mitochondria produce iron-sulphur clusters and haem groups required for many mitochondrial and cytosolic enzymes, including all four complexes of the respiratory chain. Therefore loss of iron could disrupt respiration.” Although our data does clearly indicate that chelation of iron is responsible for inducing mitophagy we acknowledge that we cannot completely rule out that iron chelators could be inducing mitophagy through chelation of another metal.

4. I do not agree with the comment on p4 "showing that iron chelation is specifically inducing mitochondrial turnover rather than general autophagy". A general autophagy inducer (starvation, TOR inhibition) surely would also lead to the increase in COXIV-LC3 co-localising puncta in a nonspecific manner, but perhaps in comparison DFP induces a greater proportion of co-localising puncta. In any case, this would be an relevant comparison to make given that a main claim is that the effect is specifically inducing mitophagy. I appreciate that a TOR inhibitor (Ku-00063794) is described here but a more 'typically' one (rapamycin/Torin1) should be used to compare (see comment below).

As suggested, to help clarify this point we have carried out a comparison of amino acid starvation (EBSS)-induced autophagy and DFP-induced autophagy. We found that 2h EBSS treatment produced approximately the same number of autophagosomes as 24h DFP treatment (as visualized by LC3 staining). However, with EBSS only 10% of these structures co-localised with mitochondria, which was not further increased by bafilomycin. In contrast over 40% LC3-COXIV was observed with DFP, which was further increased with bafilomycin. Therefore our interpretation is that there is something specific about DFP-induced autophagy that is leading to mitochondrial turnover when compared to a similar level of autophagy induced by EBSS treatment. We do note that DFP-induced autophagy is unlikely to be exclusively mitophagy, but as of yet we do not know what the other targets are. We have added the following text to describe the new data shown below.

From Results and Discussion, Loss of iron induces mitophagy:

“Immunofluorescence showed DFP treatment caused an increase in LC3 puncta (autophagosome) formation in SH-SY5Y cells; ~45% of these structures co-localised with COXIV, a complex IV component (Fig. 2A-B). Importantly, this colocalisation increased to 65% after bafilomycin addition. This contrasts to starvation-induced autophagy, that while inducing a similar number of autophagosomes (Fig. S2A), did not result in significant LC3-COXIV co-localization (Fig. 2A-B). Thus iron chelation specifically induces mitophagy rather than general autophagy.”

Figure 2.

Figure S2.

We apologise for not including information about the KU-0063794 compound. The compound was developed by KUDOS and then AstraZeneca and appears to be a very specific mTOR active site binder. Its cell-based effects are close to that of Torin1. The compound was closely analysed and published by another lab in our department. We include that reference in the supplementary methods section (García-Martínez JM et al. Ku-0063794 is a specific inhibitor of the mammalian target of rapamycin (mTOR). *Biochem J.* 2009 Jun 12;421(1):29-42). It is also referenced in the Autophagy Guidelines paper (Klionsky et al. *Autophagy* 2012, 8: 445-544). As suggested by the

reviewer we also screened rapamycin (now included in Fig. 1C) and found it to be similar to the KU-0063794 compound in its lack of mitophagy inducing ability. The supplier of KU-0063794 is now also included.

From Supplementary Methods:

"KU-0063794 and A-769662 were purchased from Tocris Biosciences (Bristol, UK), the specificity of KU-0063794 as an mTOR inhibitor has been previously determined [1] and it is a recommended autophagy agonist [2]."

5. The nature of the Parkinson's patient's mutation is not clear. Is 255delA nucleotide? 255A doesn't correspond to protein sequence.

We apologise for the ambiguity, the mutation is a nucleotide deletion that results in a premature stop-codon. The text has been changed to clarify this.

From Results and Discussion, Mitophagy is independent of PINK1 and Parkin:

"As a final confirmation, we obtained primary human dermal fibroblasts from a healthy individual and from a patient suffering from early-onset Parkinson's disease, attributed to mutations in the *PARK2* gene that codes for Parkin (compound heterozygous for a 255delA nucleotide deletion causing a premature truncation and an EXON 3-4 deletion)."

6. It is striking that not a single statistical test has been done. Have the authors verified that there effects are so robust that this is not necessary?

We apologise for the absence of statistical analysis in the original submission. All data has now been analysed, using either Student's T-test or ANOVA followed by Fisher's Least Significant Difference test. Where a difference in observation was inferred in the original submission, statistical analysis has been carried out to show significance with a p-value of 0.05 or lower. This analysis is included in all figures and a description of the tests are given in the Supplemental Methods section.

From Supplementary Methods:

"Statistical Analysis

Statistical significance was determined using unpaired Student's t-test for single comparisons (Figures 2D, 2F, 2H, 3C, 4D, 4E, 4G, S2A) and for multiple treatments ANOVA was performed followed by Fisher's Least Significant Difference Test for individual comparisons (Figures 1C, 1E, 1G, 2B, 2E, 3D, 3E, 3F, 4B, 4I, S1D, S1E, S1F, S3A, S3B, S3D, S4D). In all cases individual comparisons were made to the control condition and $P < 0.05$ was considered significant. Data was analysed using GraphPad Prism 6 software (GraphPad Software Inc. La Jolla, CA, USA)."

Minor:

7. p4. "Additionally, two other structurally distinct iron chelators (deferoxamine and Dp44mT) stimulated mitophagy (Fig. S3)". This should refer to Fig. S2.

This has been changed, but the supplemental data order has also been altered to fit the revised manuscript.

8. Also, one presumes deferoxamine is what is abbreviated as DFO. This should be stated in the text.

The reviewer is right and we apologise for not stating this. The first mention of deferoxamine in the text is on page 3 and we follow this with (DFO).

9. Fig. 2 legend. I don't understand what the "[39][39][38][41]" and "[38][38][37][37]" are. It doesn't fit with the references (especially as there isn't a ref 41). Maybe these are typos?

Sorry, these are typos and have been removed.

10. Fig. S2. CoCl₂ and DMOG are not discussed. Some reference should be made to the rationale for their inclusion in the results or should be removed.

We apologise for the omission. Both compounds inhibit the prolyl-hydroxylase responsible for HIF1 degradation (CoCl₂ displaces its iron co-factor and DMOG is an analog of another co-factor oxoglutarate) and are commonly used as chemical hypoxia mimetics. We have added this to the text

From Results and Discussion, Effects of iron chelation on mitochondrial function:

"Iron chelation is known to stabilise the oxygen responsive transcription factor HIF1 α , through inhibition of the proline hydroxylase involved in its degradation. Hypoxia has also been shown to induce mitophagy via HIF1 [17]. We tested conditions that stabilise HIF1 α (including hypoxia, proline hydroxylase inhibitors such as DMOG and CoCl₂, and iron chelation) with our assay and found they also induce mitophagy (Fig. S3A-B)."

11. Similarly, I am not familiar with the efficacy of Ku-00063794 which is not cited nor the source given.

See above for point 4. We now cite and give source (Tocris) in Methods.

12. The cell culture conditions for growth under galactose vs glucose medium was not clear.

A section has been added to supplemental methods detailing the conditions for this experiment.

From Supplementary Methods:

"Galactose Incubation

All cells were washed twice with glucose-free DMEM with 10mM galactose + 10% FBS. Cells grown under glucose were then incubated in DMEM containing 25mM glucose + 10% FBS and cells grown under galactose were incubated in glucose-free DMEM containing 10mM galactose + 10% FBS for 48 hours prior and during treatment with 1mM DFP as described above."

Referee #3:

This is an interesting report by Allen and colleagues who have developed a dual-fluorescence-based assay to track mitochondrial engulfment by the lysosome. Importantly, they have used it to demonstrate that iron chelation promotes mitophagy in a parkin/PINK1-independent manner. Although the concept of the tandem fluorescence autophagy reporter is not new, it is rather clever of the authors to apply this technique to assay for mitophagy. Overall, the main finding is novel notwithstanding the lack of mechanistic insights on how the chelation of iron results in mitochondrial clearance.

I do have a number of comments/suggestions that might help the authors to improve the manuscript further, as listed below.

We would like to thank the reviewer for their time and helpful comments.

1. A consistent problem with the quantitative data provided in this paper is the lack of statistical

evaluation. Please provide statistics for all the bar-graphs presented.

We apologise for the absence of statistical analysis in the original submission. All data has now been analysed, using either Student's T-test or ANOVA followed by Fisher's Least Significant Difference test. Where a difference in observation was inferred in the original submission, statistical analysis has been carried out to show significance with a p-value of 0.05 or lower. This analysis is included in all figures and a description of the tests are given in the Supplemental Methods section.

From Supplementary Methods:

“Statistical Analysis

Statistical significance was determined using unpaired Student's t-test for single comparisons (Figures 2D, 2F, 2H, 3C, 4D, 4E, 4G, S2A) and for multiple treatments ANOVA was performed followed by Fisher's Least Significant Difference Test for individual comparisons (Figures 1C, 1E, 1G, 2B, 2E, 3D, 3E, 3F, 4B, 4I, S1D, S1E, S1F, S3A, S3B, S3D, S4D). In all cases individual comparisons were made to the control condition and $P < 0.05$ was considered significant. Data was analysed using GraphPad Prism 6 software (GraphPad Software Inc. La Jolla, CA, USA).”

2. In characterizing the utility of the mCherry-GFP-Fis1 reporter, it will be informative to supplement the time course study shown in Fig. 1F with images showing the localization of the reporter in cells treated with DFP to examine the progressive changes that take place with time. Although the authors stated that DFP-mediated mitophagy does not involve global changes in mitochondrial localization, they nonetheless noted that those undergoing mitophagy were separated from the network.

As suggested we have now included images of the mitochondrial network (visualised with our mitophagy tag) following the time course of DFP treatment. This data is shown in Fig S1F, along with the original panel from the previous Fig. 1F).

3. The authors would need to clarify the status of endogenous parkin in U2OS osteosarcoma cells before stating that their "assay is sensitive enough to observe parkin-dependent mitophagy under endogenous parkin conditions (p.3)". (Many cancer cell lines are deficient in parkin expression). Importantly, are the DFP-induced mitochondrial puncta positive for parkin or PINK1 in these cases?

We appreciate the reviewer's concern and based on this and comments from the other reviewers we have carried out additional experiments to analyse the role of PINK1 and Parkin in depolarisation-induced mitophagy. As is mentioned in the response to Reviewer 1, we carried out PINK1 siRNA in neuroblastoma cells and although this was sufficient to block MFN2 degradation it had no effect on mitophagy. Additionally we found that CCCP could stimulate mitophagy in the primary Parkinson's fibroblasts, which lack Parkin expression. Taken together we infer that even though the PINK1/Parkin pathway is activated, it is not essential for the observed mitophagy. We have modified the text and included additional data in Fig. 4 as shown above in the response to

reviewer 1 (point 3).

We did though test to see if mitochondria from cells treated with DFP were positive for PINK1. Treatment with oligomycin/antimycin led to a large increase in PINK1 staining in very close proximity to ATP synthase staining. In contrast, DFP did not. This data is shown in Fig. S4B (and below). This further supports the notion that DFP-induced mitophagy is independent of the PINK1/Parkin pathway.

4. Fig. 2D: Control (instead of control + Baf) would be more useful here to illustrate the depletion of mitochondrial-localized proteins in the presence of DFP and their impaired degradation in the presence of DFP + Baf. Further, it would be beneficial to include an additional lane to show the effect of proteasome inhibition on the level of the proteins examined (especially Mfn2) to support the author's suggestion that Mfn2 is degraded by the proteasome in a DFP-dependent manner. This is important to clarify as Mfn2 degradation is thought to be initiated by parkin-mediated ubiquitination, which is apparently not essential here.

For Figure 2D, we had removed the control bar for each sample to save space on what is already a crowded figure. For each protein the samples have been normalized to tubulin and are given as a percentage of the control condition and so each control condition value is 100% (this was so the different proteins could all be compared on one chart). We apologise that this was not clear and have now added a dashed line to indicate where the control levels are. If the reviewer feels strongly about this, then we will happily add the control bars back. We have modified figure 2 legend to indicate this.

From Figure 2 legend:

“C. Example immunoblot and D. Quantitation of mitochondrial proteins and autophagy markers in SH-SY5Y cells treated with 1mM DFP for 24h, 50nM Bafilomycin A1 (Baf) was added for final 16h of treatment. Data relative to control condition - dotted line represents control value (100%).”

Degradation of MFN2 does indeed appear to be by the proteasome as we can rescue levels by treatment with bortezomib. This data is now shown in Fig S2B (and below). We have also modified the text to indicate this.

From Results and Discussion, Loss of iron induces mitophagy:

“Immunoblotting showed protein from each mitochondrial compartment decreased ~50% following DFP treatment (Fig. 2C-D). This was prevented with bafilomycin indicating the decrease was dependent on lysosomal degradation. The exception was mitofusin2 (MFN2) that is degraded via the proteasome prior to mitophagy ([16] and Fig. S2B).”

As mentioned by the reviewer, loss of MFN2 appears to be dependent on PINK/Parkin. However, our data from the siRNA of PINK1 suggests this is not essential for mitophagy (figure 4C-E).

5. Curiously, the level of p62 appears to increase in DFP treated cells relative to control alone (Fig. 2D). This begs the question on whether DFP-induced mitophagy requires p62, which plays a contentious role in the parkin/PINK1 mitophagy pathway. Related to this, is K63-linked ubiquitination involved here, as parkin is thought to perform this mode of ubiquitination in preparing the damaged organelle for clearance by mitophagy?

We agree with the reviewer that the p62 data could imply its involvement. We are trying to delve deeper into the mechanism of iron chelation induced mitophagy and hope to publish the role of ubiquitination (and p62) in a follow-up study. We have modified the text on p62:

From Results and Discussion, Loss of iron induces mitophagy:

“Expression of the autophagy adaptor and substrate SQSTM1/p62 increased following mitophagy induction and the amount of p62 was further enhanced with bafilomycin treatment. This suggests a role for p62 in DFP-induced mitophagy, though work is needed to clarify this.”

6. It is curious to note in Fig. 4A that despite the abundance of full length PINK1 (which serves to recruit parkin) in cells treated with either CCCP or oligomycin/antimycin, the expression of parkin is significantly reduced. The authors suggested that parkin in this case is degraded by the proteasome, which is easily testable using proteasome inhibitor treatment.

This result has been previously published and referenced in the manuscript (Rakovic A, et al. *J Biol Chem.* 2013 Jan 25;288(4):2223-37) and as suggested we added bortezomib to try and rescue Parkin levels. We found bortezomib partially restored Parkin levels in response to oligomycin/antimycin, but long-term bortezomib treatment in general reduced Parkin levels without additional treatments, making it difficult to attribute all of the Parkin loss to the proteasome. We have added the following text to the manuscript.

From Results and Discussion, Mitophagy is independent of PINK1 and Parkin:

“We also noted loss of Parkin upon CCCP or oligomycin/antimycin treatment (Fig. 4A-B), which correlates with previous observations suggesting Parkin activation leads to autoubiquitylation and increased proteasomal turnover [8]. In support, we find this loss is partially rescued with bortezomib (Fig. S4C).”

7. Show anti-parkin staining in Fig. 4C.

Due to the reduction in total Parkin levels upon oligomycin/antimycin or CCCP treatment we are unable to detect endogenous Parkin on mitochondria. However, we are able to detect endogenous PINK1 as mentioned and show in the above response to comment 3. A limitation of the tandem-tag assay is that when cells are permeabilized for antibody staining (e.g. Parkin), the pH gradient in the lysosome is disrupted and the GFP signal is restored and therefore red-alone puncta are lost. Therefore we were unable to co-stain red-alone puncta with antibodies such as parkin.

8. Considering the current debate surrounding the role of parkin/PINK1 pathway in neurons, it would be interesting to examine if DFP induces mitophagy in primary neurons using the author's tandem reporter assay (particularly in view of the author's proposal that iron accumulation in AD or PD brains may compromise mitophagy).

This would be an interesting and important study to conduct. Unfortunately we do not currently have the capability to carry out such studies and would need to establish an outside collaboration.

2nd Editorial Decision

27 September 2013

Thank you for your patience while we have reviewed your revised manuscript. As you will see from the reports below, the referees are now all positive about its publication in EMBO reports, although referee two asks for some minor text changes, which can be easily implemented. I am therefore writing with an 'accept in principle' decision, which means that I will be happy to accept your manuscript for publication once these issues have been attended to. Please also move the description of the statistical analysis and mitophagy assay quantification to the main Materials and Methods section.

We now encourage the publication of original source data -particularly for electrophoretic gels and blots, but also for graphs- with the aim of making primary data more accessible and transparent to the reader. If you agree, you would need to provide one PDF file per figure that contains the original, uncropped and unprocessed scans of all or key gels used in the figures and an Excel sheet or similar with the data behind the graphs. The files should be labeled with the appropriate figure/panel number, and the gels should have molecular weight markers; further annotation could be useful but is not essential. The source files will be published online with the article as supplementary "Source Data" files and should be uploaded when you submit your final version. If you have any questions regarding this please contact me.

Finally, as a standard procedure, we edit the abstract of manuscripts to make them more accessible to a general readership. Please find the edited abstract, which you will see addresses one of the points raised by referee 2, at the end of this email and let me know if you do NOT agree with any of the changes.

If all remaining issues have been attended to, you will then receive an official decision letter from the journal accepting your manuscript for publication in the next available issue of EMBO reports. This letter will also include details of the further steps you need to take for the prompt inclusion of

your manuscript in our next available issue.

Thank you for your contribution to EMBO reports.

Edited title and abstract

In this study, we develop a simple assay to identify mitophagy inducers based on the use of fluorescently-tagged mitochondria that undergo a colour change upon lysosomal delivery. Using this assay, we identify iron chelators as a family of compounds that generate a strong mitophagy response. Iron-chelation-induced mitophagy requires that cells undergo glycolysis, but does not require PINK1 stabilization or Parkin activation, and occurs in primary human fibroblasts as well as those isolated from a Parkinson's patient with Parkin mutations. Thus, we have identified and characterised a mitophagy pathway, the induction of which could prove beneficial as a potential therapy for several neurodegenerative diseases in which mitochondrial clearance is advantageous.

REFEREE REPORTS:

Referee #1 (Report):

The author's have adequately addressed my criticisms and I now feel that their manuscript is worthy of publication.

Referee #2 (Report):

The revised manuscript from Allen and colleagues is greatly improved from the original, and in my opinion the authors have done a good job of addressing the criticisms raised. The additional data, as well as the clarifications, have greatly strengthened the story. I have a few remaining minor comments that reflect the wording of specific claims in the text, which should be addressed, but otherwise I recommend for publication.

Abstract: "Iron chelation induced mitophagy required glycolytic cells". This phrase is rather clunky. I'm not sure cells can be described as 'glycolytic' or otherwise since they are all (presumably) capable of glycolysis and OXPHOS.

p4. "We tested ... DMOG and CoCl₂ ... and found they also induce mitophagy (Fig. S3A-B)", only DMOG doesn't (N.S.). Please amend.
But also DMOG is 'N.S.' for HIF1a/BNIP3 protein level change, so likely it wasn't working properly.

p5. "as the autophagy is specific for mitochondria". As raised in the rebuttal, the authors acknowledge that the effect of DFP is "unlikely to be exclusively mitophagy", I would suggest amending the above statement which still implies that the authors think the effect is 'specific' for mitophagy. Perhaps 'selective' is a less absolutely term.

p5. "This infers that". Incorrect use of 'infer'. While you can infer, the data can't. Suggested changes; 'This indicates/suggests ...' or 'We infer...'

The new data showing the failure to prevent O+A or CCCP-induced mitophagy by PINK1 siRNA is interesting and reflects the growing complexity of this field. However, I would encourage the authors to interpret this a bit more cautiously. The 4C blot clearly shows some remnant of PINK1 after siRNA, which may be sufficient to still induce mitophagy.

Referee #3 (Report):

In their revised version, the authors have addressed the majority of my concerns, which I am overall happy with. Although the mechanistic details regarding how chelation of iron results in mitochondrial clearance remain unclear, the report in its current form is sufficiently interesting and novel and also important to our understanding of the dynamics of mitophagy.

2nd Revision - authors' response

30 September 2013

We are thrilled with your decision to accept our manuscript in principle for publication in EMBO Reports and would like to thank your efforts, and that of the reviewers, in this process.

We are happy with the changes to the abstract and have moved the mitophagy assay details and statistical analysis to the main Materials and Methods section. We have made the following textual changes as suggested by Reviewer 2 (see below) and also uploaded the source western blot data and the associated quantitation.

Response to Reviewer 2:

- p4. "We tested ... DMOG and CoCl₂ ... and found they also induce mitophagy (Fig. S3A-B)", only DMOG doesn't (N.S.). Please amend.
But also DMOG is 'N.S.' for HIF1a/BNIP3 protein level change, so likely it wasn't working properly.

"We tested conditions previously reported to stabilise HIF1 α (including hypoxia, proline hydroxylase inhibitors such as DMOG and CoCl₂, and iron chelation) with our assay. All conditions that significantly stabilised HIF1 α also induced mitophagy (Fig. S3A-B)."

- p5. "as the autophagy is specific for mitochondria". As raised in the rebuttal, the authors acknowledge that the effect of DFP is "unlikely to be exclusively mitophagy", I would suggest amending the above statement which still implies that the authors think the effect is 'specific' for mitophagy. Perhaps 'selective' is a less absolutely term.

"We reasoned as the autophagy is selective for mitochondria, iron loss may impair mitochondrial function that in turn signals for mitophagy."

- p5. "This infers that". Incorrect use of 'infer'. While you can infer, the data can't. Suggested changes; 'This indicates/suggests ...' or 'We infer...'

"This indicates that mitophagy induction by oligomycin/antimycin or CCCP is potentially different from iron chelation."

- The new data showing the failure to prevent O+A or CCCP-induced mitophagy by PINK1 siRNA is interesting and reflects the growing complexity of this field. However, I would encourage the authors to interpret this a bit more cautiously. The 4C blot clearly shows some remnant of PINK1 after siRNA, which may be sufficient to still induce mitophagy.

On p6 we added a sentence: "Surprisingly, loss of PINK1 had no effect on CCCP or oligomycin/antimycin-induced mitophagy suggesting PINK1, and MFN2 turnover, is dispensable for mitophagy. However, the remnant of PINK1 remaining following siRNA could be sufficient to induce mitophagy."

I am very pleased to accept your manuscript for publication in the next available issue of EMBO reports. Thank you for your contribution to our journal.

As part of the EMBO publication's Transparent Editorial Process, EMBO reports publishes online a Review Process File to accompany accepted manuscripts. As you are aware, this File will be published in conjunction with your paper and will include the referee reports, your point-by-point response and all pertinent correspondence relating to the manuscript.

If you do NOT want this File to be published, please inform the editorial office within 2 days, if you have not done so already, otherwise the File will be published by default [contact: emboreports@embo.org]. If you do opt out, the Review Process File link will point to the following statement: "No Review Process File is available with this article, as the authors have chosen not to make the review process public in this case."

Thank you again for your contribution to EMBO reports and congratulations on a successful publication. Please consider us again in the future for your most exciting work.